# UNSUPERVISED LEARNING OF OBJECT-CENTRIC REPRESENTATION FROM MULTI-VIEWPOINT SCENES

## ABSTRACT

Objects in a 2D image are influenced by factors like perspective, illumination, and occlusion in the corresponding 3D scene. This results in the challenge of identifying objects across different viewpoints. Humans can effortlessly identify objects from different viewpoints by recognizing their invariant characteristics in 3D dimensions. Motivated by this observation, we propose an object-centric learning method named **L**earning **O**bject-centric **R**epresentation from **M**ulti-viewpoint (LORM), which learns the representations of objects from multi-viewpoint scenes *without any supervision*. LORM leverages a novel slot attention encoder to decompose the representation of a scene into two distinct components: a viewpoint representation and several object representations. The former encompasses the viewpoint-dependent attributes (i.e., camera position and lighting) of the image observed from each viewpoint, while the latter captures the viewpoint-independent features (i.e., appearance, shape, scale, rotation and position) of the object across various perspectives. We propose a mixture patch decoder to enable LORM to simultaneously handle complex scenes and reconstruct an individual object's 2D appearance and shape at a specific viewpoint through the corresponding object representation and viewpoint representation. Extensive experiments are conducted on three complex simulation datasets, and the results demonstrate that our proposed method outperforms compared methods in individual object reconstruction while achieving comparable performance in scene decomposition.

## 1 INTRODUCTION

When humans observe their surrounding world, numerous objects are regarded as features to understand the world (Johnson, 2010). Compared with perceiving the whole scene directly, knowledge can be acquired from the surrounding world more efficiently through compositional perception (Fodor & Pylyshyn, 1988). Therefore, to make the artificial intelligence systems learn the knowledge of the world as efficiently as human beings, it is crucial to perceive the scene in a compositional way (Lake et al., 2017). **O**bject-**C**entric **L**earning (OCL) is a compositional scene perception method that focuses on separately learning the representations of individual objects in a scene.

Recently, object-centric representation learning has attracted much attention and multiple outstanding works have been summarized (Yuan et al., 2022a). For example, the methods, such as AIR (Eslami et al., 2016), GMIOO (Yuan et al., 2019), SPACE (Lin et al., 2019), Slot Attention (Locatello et al., 2020), GENESIS-v2 (Engelcke et al., 2021), SLATE (Singh et al., 2022a), DINOSAUR (Seitzer et al., 2023) etc., work well on the static scene. The methods, such as SQAIR (Kosiorek et al., 2018), SCALOR (Jiang et al., 2019), G-SWM (Lin et al., 2020), etc., learn better compositional scene representations by modeling the motions and relationships of objects in the scene. The methods, such as SIMONe (Kabra et al., 2021), MulMON (Li et al., 2020), OCLOC (Yuan et al., 2022b), focus on observing the scene from multiple viewpoints. The methods, such as SAVi (Kipf et al., 2022), SAVi++ (Elsayed et al., 2022), STEVE (Singh et al., 2022b), apply object-centric learning to videos. Despite the proficiency of the above methods in extracting object-centric representations, they still have some limitations in processing complex scenes. On the one hand, methods such as Slot Attention, OCLOC, and SIMONe cannot segment the object in the complex scenes well. On the other hand, methods such as DINOSAUR and STEVE face difficulties constructing individual object images through their corresponding representations to obtain the gratifying performance of decomposition on complex scenes.

A fundamental ability of the human brain is invariant object recognition, which involves the rapid and precise identification of objects based on their relatively consistent (invariant) features despite variances in size, rotation, and position (Karimi-Rouzbahani et al., 2017). What's more, humans have the ability to distinguish between relatively consistent (invariant) features and view-specific (non-invariant) features of an object from different viewpoints (Turnbull et al., 1997). Inspired by these human cognitive capabilities, we propose learning the viewpoint-independent representation of objects in 3D scenes to identify consistent objects in the image viewed from multiple perspectives.

In this paper, we propose a novel object-centric learning method, called **L**earning **O**bject-centric **R**epresentation from **M**ulti-viewpoint (LORM), for learning object-centric representation from multiple-viewpoint scenes in an *unsupervised* manner. Specifically, we assume that a compositional scene representation consists of viewpoint and multiple object representations. Viewpoint representations correspond to the scene's global view-specific elements (such as camera position). Object representations indicate objects' viewpoint-independent attributes (such as appearance, shape, scale and position) in 3D scenes. The former is the output of a viewpoint encoder, and the latter is obtained by a slot attention encoder with the image feature and viewpoint representation as inputs. Moreover, we propose a mixture patch decoder to reconstruct the individual object image observed from a specific viewpoint by the corresponding object and viewpoint representations. It can not only decompose complex scenes but also reconstruct the individual object image.

The experiment section uses three complex simulations (i.e., CLEVR-A (Johnson et al., 2017), SHOP (Nazarczuk & Mikolajczyk, 2020) and GSO (Greff et al., 2022)) multi-view scene data to evaluate the proposed method. Two representative multi-viewpoint-based methods, OCLOC (Yuan et al., 2022b) and SIMONe (Kabra et al., 2021), and two with outstanding performance in complex scenes, DINOSAUR (Seitzer et al., 2023) and STEVE (Singh et al., 2022b), are selected as comparison methods. The abundant experimental results show that the proposed method not only has outstanding performance of decomposition on complex scenes but also can reconstruct the individual object image well.

In summary, the contributions of this work are as follows:

1) We propose a novel object-centric representation learning method that learns object representation from multi-view scenes without supervision.

2) We propose an object-centric encoder, consisting of a viewpoint encoder and slot attention encoder, to disentangle the scene representation into viewpoint and object representation.

3) We propose a mixture patch decoder to reconstruct an object image with the individual object and corresponding viewpoint representations as inputs.

4) LORM is the first object-centric learning method that can decompose complex scenes and reconstruct the image of individual objects simultaneously.

## 2 RELATED WORKS

In recent years, a large number of object-centric representation learning methods have been proposed to learn compositional scene representations, which are a collection of representations of objects in the scene. According to the form of the input image, the current methods can be roughly divided into three categories: Single-image-based, Video-based and Multi-view-based.

**Single-image-based:** N-EM (Greff et al., 2017) and AIR (Eslami et al., 2016) are two of the earlier representative methods. The former initializes the representations of all objects and iteratively updates them, while the latter extracts the representations of objects in sequence according to the attention mechanism. GMIOO (Yuan et al., 2019) improvs N-EM and AIR by handling occlusions between objects and models objects and the background separately. MONet (Burgess et al., 2019) uses the U-Net(Ronneberger et al., 2015) to predict the masks of objects, which are used to serially extract the representations of objects via a variational autoencoder (Kingma & Welling, 2013) with concatenated the scene image as input. IODINE (Greff et al., 2019) utilizes iterative variational inference to learn the representation of objects in the scene. SPACE (Lin et al., 2019) can effectively handle scenes with complex backgrounds and a large number of foreground objects by modeling the background with spatial mixture models and extracting the representations of foreground objects with parallel spatial attention. GNM (Jiang & Ahn, 2020) models the layout of the scene, while

GENESIS (Engelcke et al., 2019) models the interrelationship between objects in an autoregressive manner. GENESIS-V2 (Engelcke et al., 2021) improves the performance of GENESIS by predicting mask attention of objects with the Instance Colouring Stick-Breaking Process. Slot Attention (Locatello et al., 2020) first initializes the representations of the object and iteratively updates them according to the similarity between the representations and the local features of the scene image. DINOSAUR (Seitzer et al., 2023) uses a pre-trained self-supervised transformer (Caron et al., 2021) to handle real-world scenes. DINOSAUR is similar to our work in that it reconstructs patch features. However, DINOSAUR cannot reconstruct the individual objects and the whole scene.

**Video-based:** Rational-NEM (van Steenkiste et al., 2018) discovers objects from video scenes and learns the physical interactions between them in an unsupervised manner. SQAIR (Kosiorek et al., 2018) extends the discovery and propagation modules in AIR to discover and track objects throughout the sequence of frames. In addition, SQAIR can generate future frames conditioning on the motion of objects in the current frame. SCALOR (Jiang et al., 2019) proposes a spatially parallel attention and proposal-rejection mechanism to focus on the learning of compositional scene representations for the scene with a large number of objects. G-SWM (Lin et al., 2020) unifies the critical attributes of previous models in a framework of principles and proposes two crucial new abilities: multimodal uncertainty and situation awareness. SAVi (Kipf et al., 2022) and SAVi++ (Elsayed et al., 2022) use optical flow supervision to learn temporal information between object representations of adjacent frames. Based on the transformer decoder, STEVE (Singh et al., 2022b) makes significant improvements on various complex and naturalistic videos. Similar to STEVE, our work also uses a Discrete VAE (DVAE)(Im Im et al., 2017)to reconstruct the whole scene. The difference with STEVE is that our method can reconstruct the image of the individual object.

**Multi-view-based:** MulMON (Li et al., 2020) is the method that first learns the compositional representation of a multi-object scene by leveraging multiple views information and iteratively updating object representations in a scene over multiple viewpoints based on IODINE. SIMONe (Kabra et al., 2021) learns the frame latent variables that capture time-varying information and object latent variables that are invariant and time-independent, respectively, with the frames in the sequence as input. ROOTS (Chen et al., 2021) divides a 3-dimensional scene int grid cells extending SPACE (Lin et al., 2019) and then estimates the size and position of each object in 3-dimensional space. The sizes and positions of objects in 3-dimensional grids are converted to the 2-dimensional coordinates in each viewpoint image according to annotated viewpoints. The representation of each object is obtained by encoding the features of objects over multiple viewpoints. Inspired by humans' ability for so-called 'object constancy', OCLOC (Yuan et al., 2022b) learns the representation of 3D objects from a scene with multiple unspecified viewpoints by disentangling viewpoint-dependent and viewpoint-independent latent variables. Like SIMONe and OCLOC, our method learns the disentangled representation of viewpoint and object from multi-view scenes through an unsupervised approach. The performance of SIMONe and OCLOC in complex scenes is unsatisfactory.

## 3 METHOD

The proposed LORM comprises three components: (1) Object-Centric Encoder, which extracts viewpoint and object representations; (2) Mixture Patch Decoder, which reconstructs the entire scene and the image of individual objects; and (3) Image Encoder-Decoder, which encodes the scene into a single representation and converts it into the whole scene via a decoder. The overview of LORM is shown in Figure 1.

### 3.1 OBJECT-CENTRIC ENCODER

A visual scene is assumed to consist of multiple objects and can be observed from $V$ different viewpoints. $K$ is the maximum number of objects that can appear in the scene. Given images $\boldsymbol{x}_{1:V}$ of the 3D scene randomly observed from $V$ viewpoints, LORM can extract $V$ viewpoint representations and $K$ object representations by the Objec-Centric Encoder which includes View Encoder $f_{\text{enc}}^{\text{view}}$ and Slot Attention Encoder $f_{\text{enc}}^{\text{sa}}$. $f_{\text{enc}}^{\text{view}}$ consists of a convolutional neural network and two layers of fully connected networks, which extract viewpoint-dependent attributes representation $\boldsymbol{s}_v^{\text{view}}(1 \leq v \leq V)$ that is the same for all objects from each viewpoint image $\boldsymbol{x}_v(1 \leq v \leq V)$. Unlike existing object-centric learning methods, the Slot Attention Encoder $f_{\text{enc}}^{\text{sa}}$ in this work is input the multiple viewpoint representations $\boldsymbol{s}_{1:V}^{\text{view}}$ and the corresponding multi-view images $\boldsymbol{x}_{1:V}$ together to obtain $K$ object rep-

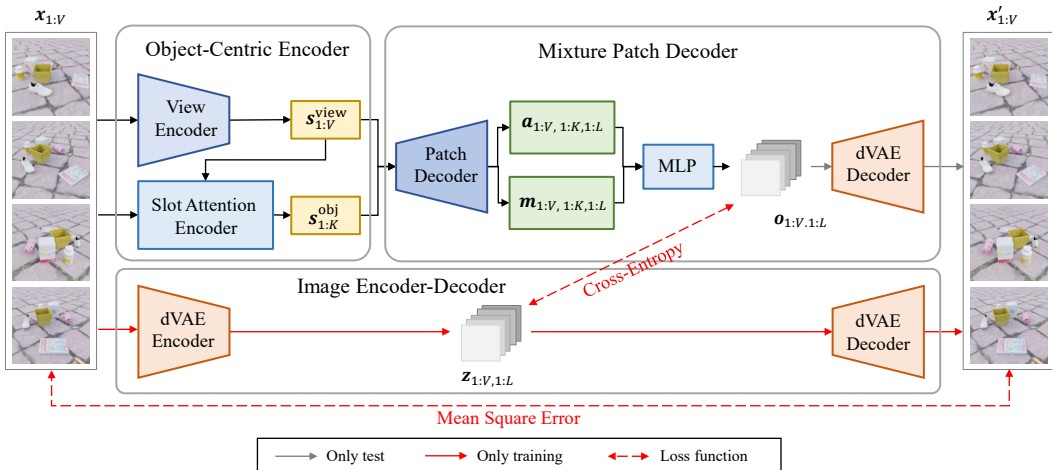

Figure 1: The overview of LORM. LORM consists of three parts:(1) Object-Centric Encoder, which converts the multi-view images into two disentanglement representations of viewpoint-dependent $s_{1:V}^{\text{view}}$ and viewpoint-independent $s_{1:K}^{\text{obj}}$; (2) Mixture Patch Decoder, which mixes each patch of each viewpoint image by weighted summing $a_{v,1:K,l}$ with $m_{v,1:K,l}$ as weights; (3) Image Encoder-Decoder, which encodes and decodes images from each viewpoint as a whole and is used to calculate the reconstruction loss only during the training process.

resentations $s_{1:K}^{\text{obj}}$ containing viewpoint-independent attributes. The process of extracting viewpoint representation and object representation can be expressed as follows:

$$s_v^{\text{view}} = f_{\text{enc}}^{\text{view}}(x_v), 1 \le v \le V, \qquad s_{1:K}^{\text{obj}} = f_{\text{enc}}^{\text{view}}(x_{1:V}, s_{1:V}^{\text{view}}).$$

The detail of $f_{\text{enc}}^{\text{sa}}$ is presented in Algorithm 1. First, the features of the $v$th viewpoint image $s_v^{\text{img}}$ are extracted by a neural networks $f_{\text{enc}}^{\text{img}}$. Next, the intermediate variables $s_{1:K}^{\text{obj}}$ are initialized by sampling from two Gaussian distributions with learnable parameters ($\tilde{\mu}^{\text{obj}}$ and $\tilde{\sigma}^{\text{obj}}$) respectively, and then iteratively updated. $s_k^{\text{obj}}$ denotes the viewpoint-independent features of the $k$th object in 3D scene. In each step of the iterative updates, the full intermediate variable $s_{1:V,1:K}^{\text{full}}$ can be obtained by broadcasting and concatenating $s_{1:V}^{\text{view}}$ and $s_{1:K}^{\text{obj}}$. The attention map $\tilde{a}_{v,1:K}$ ($1 \le v \le V$) are calculated separately for each viewpoint by normalizing the similarities between the *keys* $f_{\text{key}}(s_v^{\text{img}})$ and the *quries* $f_{\text{qry}}(s_{v,1:K}^{\text{full}})$ with the temperature $\sqrt{D_{\text{key}}}$. $f_{\text{key}}$ and $f_{\text{qry}}$ are two linear transformation networks, and $D_{\text{key}}$ are the last dimension of the output of $f_{\text{key}}$. $u_{v,1:K}$ contains the information updating $s_{1:K}^{\text{obj}}$ and is measured as the weighted average of the *values* $f_{\text{val}}(s_v^{\text{img}})$ across $N$ pixels, with attention maps $tilde{a}_{v,1:K}$ ($1 \le v \le V$) as weights. $f_{\text{val}}$ is a linear transformation network. Finally, the intermediate variable $s_{v,k}^{\text{obj}}$ is updated via a Gated Recurrent Unit $f_{\text{GRU}}^{\text{upd}}$ with $s_k^{\text{obj}}$ and $u_{v,k}$ as inputs, and the updated $s_{1:V,1:K}^{\text{obj}}$ is further averaged on the viewpoint dimension, which indicates $s_k^{\text{obj}}$ captures viewpoint-independent attributes of the $k$th object.

## 3.2 MIXTURE PATCH DECODER

One of the object-centric learning methods' essential and fundamental abilities is reconstructing the image of individual objects in the scene. However, it is a massive challenge for object-centric learning methods applied to complex and naturalistic scenes. To decompose complex natural scenes, DINOSAUR reconstructs the patch features extracted by a pre-trained self-supervised transformer (Caron et al., 2021), while STEVE reconstructs the whole scenes through the DVAE decoder. Therefore, we combine the advantages of DINOSAUR and STEVE and propose a mixture patch decoder to obtain the ability to decompose complex scenes without losing the ability to reconstruct the image of the individual objects.

---

**Algorithm 1** Slot Attention Encoder

---

**Input:** Images of $V$ viewpoints $\boldsymbol{x}_{1:V}$, viewpoint representations $\boldsymbol{s}_{1:V}^{\text{view}}$
**Output:** object representations $\boldsymbol{s}_{1:K}^{\text{obj}}$
$\boldsymbol{s}_v^{\text{img}} = f_{\text{enc}}^{\text{img}}(\boldsymbol{x}_v), \forall 1 \le v \le V$
$\boldsymbol{s}_k^{\text{obj}} \sim \mathcal{N}\big(\tilde{\boldsymbol{\mu}}^{\text{obj}}, \text{diag}(\tilde{\boldsymbol{\sigma}}^{\text{obj}})^2\big), \forall 1 \le k \le K$
**for** $t \leftarrow 1$ **to** $T$ **do** $\{\forall 1 \le v \le V, 1 \le k \le K$ in the loop$\}$
$\quad \boldsymbol{s}_{v,k}^{\text{full}} \leftarrow \big[\boldsymbol{s}_v^{\text{view}}, \boldsymbol{s}_k^{\text{obj}}\big]$
$\quad \tilde{\boldsymbol{a}}_{v,k} \leftarrow \text{Softmax}_K\Big(\big(f_{\text{key}}(\boldsymbol{s}_v^{\text{img}}) \cdot f_{\text{qry}}(\boldsymbol{s}_{v,1:K}^{\text{full}})\big)/\sqrt{D_{\text{key}}}\Big)$
$\quad \boldsymbol{u}_{v,k} \leftarrow \sum_N \text{Softmax}_N(\log \tilde{\boldsymbol{a}}_{v,k}) f_{\text{val}}(\boldsymbol{s}_v^{\text{img}})$
$\quad \boldsymbol{s}_{v,k}^{\text{upd}} \leftarrow f_{\text{GRU}}^{\text{upd}}(\boldsymbol{s}_k^{\text{obj}}, \boldsymbol{u}_{v,k})$
$\quad \boldsymbol{s}_k^{\text{obj}} \leftarrow \text{Mean}_V(\boldsymbol{s}_{1:V,k}^{\text{upd}})$
**end for**
**return** $\boldsymbol{s}_{1:K}^{\text{obj}}$

---

The viewpoint representations $\boldsymbol{s}_{1:V}^{\text{view}}$ and object representations $\boldsymbol{s}_{1:K}^{\text{obj}}$ indicate the compositional scene representations of the input 3D scene and are input into a patch decoder $g_{\text{dec}}^{\text{patch}}$ after broadcasting and concatenating. Multiple fully connected layers are used to consist of $g_{\text{dec}}^{\text{patch}}$ to reduce the complexity of the model construct as much as possible. $L$ is the number of the patch feature. The output of patch decoder $\boldsymbol{a}_{v,k,l}$ and $\boldsymbol{m}_{v,k,l}$ ($1 \le v \le V, 1 \le k \le K, 1 \le l \le L$) denote the features and mask of the $l$th patch belong to the $k$th object observed from the $v$th viewpoint. $\boldsymbol{m}_{1:V,1:K,1:L}$ is normalized in the object dimension using a softmax function. Each patch observed from each viewpoint is obtained by weighted summing the patch features $\boldsymbol{a}_{v,1:K,l}$ ($1 \le v \le V, 1 \le l \le L$) with the normalized $\boldsymbol{m}_{v,1:K,l}$ ($1 \le v \le V, 1 \le l \le L$) as weights. Each mixed patch can be converted into the log probabilities $\boldsymbol{o}_{1:V,1:L}$ for a categorical distribution with $N$ classes similar to STEVE via a multilayer perceptron $g_{\text{mlp}}$. During the training process, $\boldsymbol{o}_{1:V,1:L}$ is encouraged to approximate the output of the DVAE encoder $f_{\text{enc}}^{\text{dvae}}$ as close as possible through a cross-entropy loss. During the testing process, $\boldsymbol{o}_{v,1:L}$ is used to reconstruct the whole scene image observed from $v$th viewpoint via a DVAE decoder $g_{\text{dec}}^{\text{dvae}}$. The process of mixture patch decoder can be expressed as follows:

$$\boldsymbol{a}_{1:V,1:K,1:L}, \boldsymbol{m}_{1:V,1:K,1:L} = g_{\text{dec}}^{\text{patch}}\big(\boldsymbol{s}_{1:K}^{\text{obj}}, \boldsymbol{s}_{1:V}^{\text{view}}\big),$$
$$\hat{\boldsymbol{m}}_{v,1,l}, \cdots, \hat{\boldsymbol{m}}_{v,K,l} = \text{softmax}(\boldsymbol{m}_{v,1,l}, \cdots, \boldsymbol{m}_{v,K,l}), \qquad 1 \le v \le V, 1 \le l \le L,$$
$$\boldsymbol{o}_{v,l} = g_{\text{mlp}}\Big(\sum_{k=1}^{K} \boldsymbol{m}_{v,k,l} \cdot \boldsymbol{a}_{v,k,l}\Big), \qquad 1 \le v \le V, 1 \le l \le L,$$
$$\tilde{\boldsymbol{x}}_v = g_{\text{dec}}^{\text{dvae}}(\boldsymbol{o}_{v,1:L}), \qquad 1 \le v \le V.$$

### 3.3 IMAGE ENCODER-DECODER

Similar to STEVE, we use a Discrete VAE (Im Im et al., 2017) to encode the image $\boldsymbol{x}_v$ of the 3D scene observed from $v$th viewpoint into $L$ patches and reconstruct the whole image. The DVAE encoder network $f_{\text{enc}}^{\text{dvae}}$ can convert each image $\boldsymbol{x}_v$ into $L$ patches $\boldsymbol{z}_{v,l}(1 \le l \le L)$ indicating the log probability for a categorical distribution with $N$ classes. The whole image $\boldsymbol{x}_v$ can be constructed via DVAE decoder $g_{\text{dec}}^{\text{dvae}}$ with $\boldsymbol{z}_{v,l:L}$ as inputs.

During the training process, the reconstruction $\hat{\boldsymbol{x}}_v$ decoded from $\boldsymbol{z}_{v,l:L}$ is used to calculate the reconstruction loss according to the mean square error between $\boldsymbol{x}_v$ and $\hat{\boldsymbol{x}}_v$. $\boldsymbol{z}_{v,l:L}$ is used to calculate the cross-entropy loss with $\boldsymbol{o}_{v,l:L}$. The loss function of the proposed method can be written as

$$\mathcal{L} = \mathcal{L}_{\text{rec}} + \mathcal{L}_{\text{ce}}, \qquad \mathcal{L}_{\text{rec}} = \sum_{v=1}^{V} ||\boldsymbol{x}_v - \hat{\boldsymbol{x}}_v||_2^2, \qquad \mathcal{L}_{\text{ce}} = \sum_{v=1}^{V} \sum_{l=1}^{L} \text{Cross-Entropy}(\boldsymbol{z}_{v,l}, \boldsymbol{o}_{v,l}).$$

Table 1: Performance comparison of LORM, OCLOC, SIMONe, STEVE, and DINOSAUR in terms of all metrics on three datasets. 'SIMONe (video)' and 'STEVE (video)' denote the corresponding models that are trained and tested by inputting images observed from sequential viewpoints. Bold values indicate the best performance and underlined values indicate the next best performance. All the reported results are based on 3 times evaluations of the test sets.

| Data set | Model | AMI-A↑ | ARI-A↑ | AMI-O↑ | ARI-O↑ | mIOU↑ |
|---|---|---|---|---|---|---|
| CLEVR-A | OCLOC | 0.550±2e-3 | 0.645±2e-3 | 0.886±4e-3 | 0.893±5e-3 | 0.007±2e-3 |
| | SIMONe | 0.131±3e-5 | 0.040±2e-5 | 0.466±6e-5 | 0.372±4e-5 | 0.218±1e-4 |
| | SIMONe (video) | 0.262±3e-5 | 0.074±1e-6 | 0.909±1e-4 | 0.920±6e-5 | 0.460±2e-5 |
| | STEVE | 0.702±6e-4 | 0.824±7e-4 | 0.760±1e-3 | 0.736±2e-3 | 0.715±1e-3 |
| | STEVE (video) | **0.715±3e-4** | **0.830±7e-4** | 0.776±4e-4 | 0.759±8e-4 | **0.752±8e-4** |
| | DINOSAUR | 0.145±8e-5 | 0.010±1e-4 | **0.962±3e-4** | **0.964±9e-4** | 0.138±1e-4 |
| | LORM | 0.141±1e-4 | 0.015 ±1e-3 | 0.899±1e-3 | 0.879±1e-3 | 0.147±1e-3 |
| SHOP | OCLOC | 0.528±3e-3 | 0.656±3e-3 | 0.698±5e-3 | 0.665±8e-3 | 0.010±3e-4 |
| | SIMONe | 0.183±5e-5 | 0.070±3e-5 | 0.461±2e-4 | 0.322±1e-4 | 0.218±8e-5 |
| | SIMONe (video) | 0.368±3e-5 | 0.147±8e-6 | 0.749±6e-5 | 0.686±1e-4 | 0.475±3e-5 |
| | STEVE | 0.633±5e-4 | 0.731±6e-4 | 0.742±2e-3 | 0.757±2e- 3 | 0.651±3e-3 |
| | STEVE (video) | **0.667±9e-4** | **0.766±1e-3** | 0.781±3e-3 | 0.808±4e-3 | **0.710±4e-3** |
| | DINOSAUR | 0.302±9e-4 | 0.144±2e-3 | **0.948±8e-4** | **0.949±2e-3** | 0.338±1e-3 |
| | LORM | 0.258±1e-3 | 0.060±1e-3 | 0.904±1e-3 | 0.900±1e-3 | 0.290±1e-3 |
| GSO | OCLOC | 0.001±1e-3 | 0.001±2e-3 | 0.003±2e-3 | 0.003±3e-3 | 0.146±4e-4 |
| | SIMONe | 0.000±6e-6 | 0.000±7e-7 | 0.000±2e-5 | 0.000±3e-6 | 0.029±1e-6 |
| | SIMONe (video) | 0.000±7e-6 | 0.00±2e-6 | 0.00±2e-5 | 0.00±1e-6 | 0.056±2e-6 |
| | STEVE | 0.457±8e-4 | 0.299±1e-3 | 0.793±1e-3 | 0.801±1e-3 | 0.579±1e-3 |
| | STEVE (video) | 0.458±1e-3 | 0.300±1e-3 | 0.795±2e-3 | 0.799±3e-3 | 0.582±5e-4 |
| | DINOSAUR | **0.602±6e-4** | **0.654±5e-4** | **0.941±1e-3** | **0.954±2e-3** | **0.639±7e-4** |
| | LORM | 0.318±1e-3 | 0.131 ±1e-3 | 0.910±1e-3 | 0.918±1e-3 | 0.398±1e-3 |

## 4 EXPERIMENTS

The performance of the proposed LORM is evaluated from several aspects, including the decomposition of scenes, the reconstruction of the individual object, disentanglement learning and viewpoint learning on three complex synthetic scene datasets.

**Datasets:** Three 3D multi-object scenes with multiple viewpoints datasets are used in the experiments. CLEVR-A is an augmentation of CLEVR (Johnson et al., 2017), which is a benchmark dataset for object-centric learning methods. To make the scene more complex, we expanded the number of object categories from 3 to 10. SHOP (Nazarczuk & Mikolajczyk, 2020) is a 3D scene dataset with more complex textures and shapes than CLEVR-A, and is generated according to the official code by selecting 10 objects to make up all the scenes. GSO is a 3D scene dataset, the background and objects of which are much more complex than those of CLEVR-A and SHOP. GSO is generated by selecting 10 objects and 10 complex backgrounds to render the scene with Kubric (Greff et al., 2022). Further details of all datasets are described in the Supplementary Material.

**Comparison Methods:** Two multi-view based methods, i.e. SIMONe (Kabra et al., 2021) and OCLOC (Yuan et al., 2022b), are used to evaluate the scene decomposition performance of the proposed method. SIMONe learns object-centric representations from continuous multi-view scenes without supervision. Unlike SIMONe, OCLOC is trained unsupervised by inputting the randomly selected image from multiple viewpoints, which is similar to the proposed method. Two state-of-the-art methods for the complex and naturalistic scenes, DINOSAUR (Seitzer et al., 2023) and STEVE (Singh et al., 2022b), are also selected as comparison methods. They are used to compare the ability of the proposed method to decompose complex scenes.

**Evaluation Metrics:** Several metrics are used to evaluate the performance of scene decomposition. *Adjusted Mutual Information* (AMI) (Vinh et al., 2009), *Adjusted Rand Index* (ARI) (Hubert & Arabie, 1985), and *mean Intersection over Union* (mIOU) are used to assess the quality of segmentation. In this paper, AMI-A and ARI-A are computed considering both objects and background, while AMI-O and ARI-O are calculated considering only objects. The better the performance, the higher the value of ARI, AMI and mIOU.

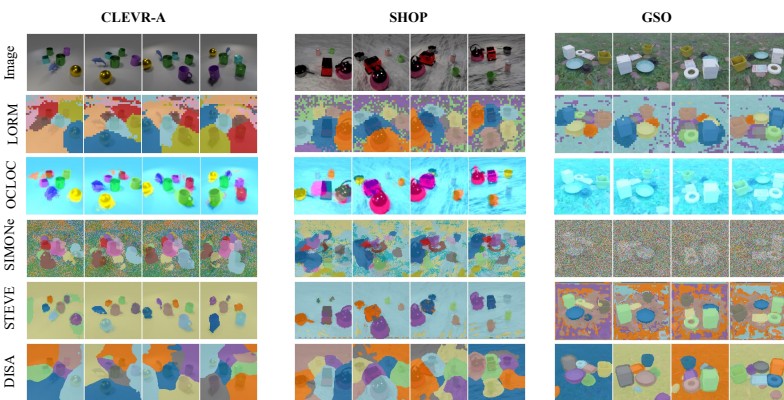

Figure 2: **The visualization of unsupervised object segmentation of LORM, OCLOC, SIMONe, STEVE and DINOSAUR on CLEVR-A, SHOP and GSO datasets.** 'Image' represents input images observed from four random viewpoints. 'DISA' is the abbreviation for DINOSAUR.

## 4.1 DECOMPOSITION OF SCENES

This section will evaluate the scene decomposition performance of LORM, OCLOC, SIMONe, STEVE and DINOSAUR from qualitative and quantitative aspects. LORM and OCLOC are trained and tested by inputting the random viewpoint images. SIMONe and STEVE are trained by inputting sequential viewpoint images and tested by random and sequential viewpoint images. DINOSAUR is trained and tested by inputting a single image.

**Quantitative results:** The comparison quantitative results of LORM, OCLOC, SIMONe, STEVE and DINOSAUR for segmentation metrics on all datasets are shown in Table 1. The segmentation metrics AMI-O and ARI-O of LORM on the most complex dataset (i.e. GSO) are significantly higher than the comparative methods (OCLOC, SIMONe and STEVE), among which the OCLOC and SIMONe methods have almost no effect on the GSO dataset. DINOSAUR performs better than LORM on all datasets due to the use of a pre-trained self-supervised Transformer, while LORM uses a simple neural network structure for end-to-end learning in an unsupervised manner. On the most complex GSO dataset, the value of AMI-O and ARI-O of LORM is slightly lower than DINOSAUR. It is worth noting that the value of AMI-O and ARI-O of LORM is much higher than STEVE on all datasets. It implies that LORM is better than STEVE at segmenting objects in complex scenes.

**Qualitative results:** We show the visualization of the segmentation performance of LORM, OCLOC, SIMONe, STEVE, and DINOSAUR on three datasets in Figure 2. In all datasets, LORM can accurately segment all objects in the scene, although the predicted object masks are a bit rough. On the contrary, STEVE can predict more befitting object masks but may miss certain objects when segmenting the scene. For example, as shown in Figure 1, the black pan is divided into the background in the segmentation map of the last viewpoint (column) image in the shop dataset. The black hat is also divided into the background in the GSO dataset. LORM can pick them out accurately. It is worth noting that SIMONe and OCLOC hardly segment foreground objects from the complex scene dataset, although they have impressive performance in CLEVR and SHOP datasets.

## 4.2 RECONSTRUCTION OF INDIVIDUAL OBJECTS

In this section, we evaluate LORM's ability to reconstruct individual objects. The visualization of the comparison of the reconstruction of the individual object between LORM, STEVE, and SIMONe is shown in Figure 3. As can be seen from Figure 3, STEVE can hardly reconstruct the corresponding images through the representation of a single foreground object on the simple scene dataset (i.e., SHOP) and the complex scene datasets (i.e., GSO). The reason may be that STEVE only focuses on reconstructing the entire scene, and during training, all object representations must be input into the decoder to reconstruct the whole scene. This causes the decoder to be unable to reconstruct when there is only a single object representation as input. On the simple scene dataset (i.e., SHOP), SIMONe can reconstruct the corresponding object image by the representation of a single foreground object. In contrast, on the complex scene dataset(i.e., GSO), SIMONe cannot

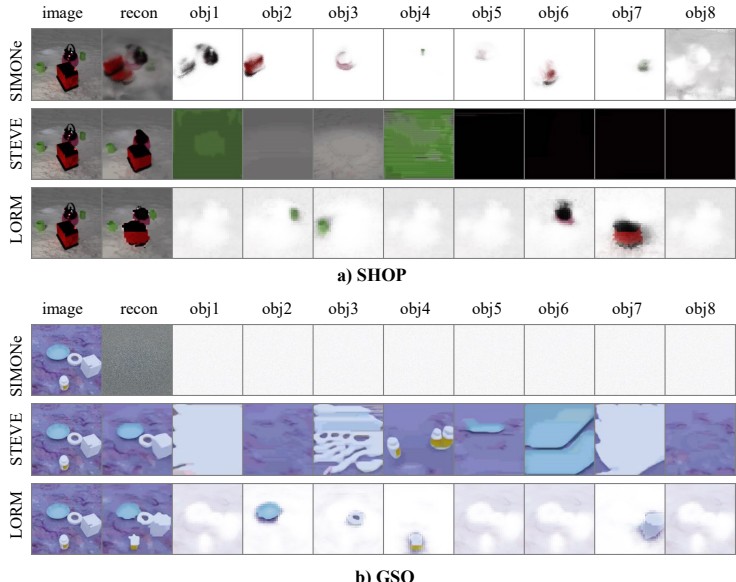

Figure 3: **The visualization of the individual object reconstruction of SIMONe, STEVE and LORM on SHOP and GSO datasets.** Figure a) shows the visualization of the individual object reconstructed by SIMONe, STEVE and LORM on the SHOP dataset. Figure b) shows the visualization of a single object reconstructed by SIMONe, STEVE and LORM on the GSO dataset.

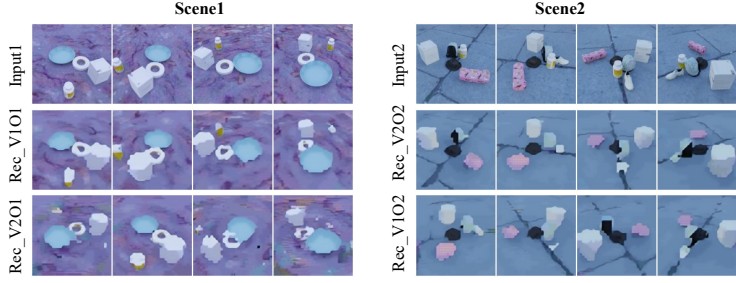

Figure 4: **The visualization of disentanglement learning of LORM on GSO dataset.** 'Input1' and 'Input2' represent two scene images observed from four random viewpoints. 'Rec_V1' and 'Rec_V2O2' represent reconstruction observed from four random viewpoints of two scenes. 'Rec_V2O1' and 'Rec_V1O2' represent, respectively, an image reconstructed using the object representation of 'Scene1' with the viewpoint representation of 'Scene2' and an image reconstructed using the object representation of 'Scene2' with the viewpoint representation of 'Scene1'.

reconstruct the object and the entire scene. The reason may be that SIMONe reconstructs the whole scene by weighted summing appearances and shapes of all objects in the scene, which results in it being too focused on reconstructing a single object and unable to process complex scene images. As shown in Figure 3, LORM can better reconstruct the corresponding image by the representation of a single object in simple and complex scenes (both SHOP and GSO datasets). LORM is the only method to handle complex scenes and reconstruct the corresponding image from a single object representation simultaneously. This is due to the proposed mixture patch decoder, which balances the impact of reconstructing the entire scene and reconstructing a single object by first reconstructing the characteristics of a single object, allowing the proposed model to handle complex scenes and reconstruct a single object image well.

## 4.3 DISENTANGLEMENT LEARNING

LORM utilizes multi-view images to improve the performance in decomposing complex 3D scenes. The multi-view images can be disentangled into viewpoint representations and object representations via the slot attention encoder of LORM. In this section, we evaluate LORM's ability to disen-

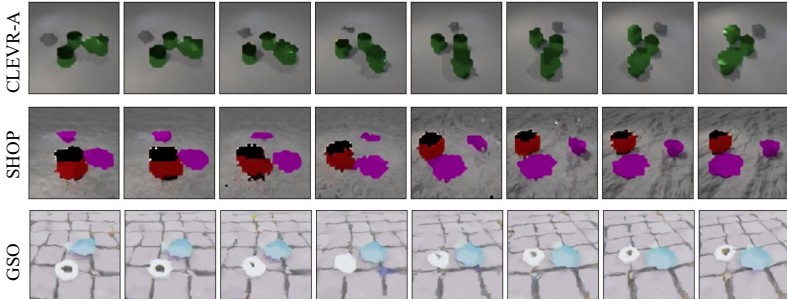

Figure 5: **The visualization of viewpoint interpolation of LORM on CLEVR-A, SHOP and GSO datasets.** Each row represents the interpolation visualization of one dataset. Interpolation between viewpoint representations of the first column image and the last column image of each row.

tangle the 3D scene attributes into viewpoint-dependent and viewpoint-independent representations according to the reconstruction decoded from exchanged viewpoint representations of two scenes. Figure 4 shows the comparison of the input images of 'Scene1' and 'Scene2' (the first row), the corresponding reconstruction (the second row), and the reconstruction after exchanging the viewpoint representation of 'Scene1' and 'Scene2' (the third row). The figure shows that the observed 2D appearance, shape, scale, and position of objects in the third-row reconstruction differ from the second-row reconstruction. In contrast, the categories and numbers of objects in the second-row reconstructions are the same as those in the second-row reconstructions. It demonstrates that the proposed model can disentangle the viewpoint and object representations in the 3D scene.

### 4.4 VIEWPOINT LEARNING

In this section, we evaluate the performance of the proposed method in learning viewpoint representation from multiple viewpoint images via an unsupervised approach. We evaluate the proposed method's viewpoint learning performance by observing a series of images constructed by decoding interpolating between the two extracted viewpoint representations. Figure 5 shows the visualization of viewpoint interpolation of the proposed method on three datasets. It can be found from Figure 5 that on the three datasets, LORM can reconstruct the scene image well by interpolating the viewpoint representation. As the viewing viewpoint changes, the objects' appearance, shape, size, and position in the reconstructed image also change accordingly. It implies that the proposed method can learn viewpoint representation well without supervision.

### 4.5 FUTURE DIRECTIONS AND LIMITATIONS

Our method is the first to achieve satisfactory complex scene segmentation capabilities without losing the ability to reconstruct individual object images. The limitation of the proposed method is that the reconstruction ability and accurate segmentation of objects need to be improved. The reason may be that the network construction of the patch decoder only uses multiple fully connected networks. In future work, the more complex neural networks will be used to replace multiple fully connected networks in the patch decoder of the proposed method. Another potential work is to use a flow-based model to encode and decode the entire scene, which is a generative model that can reversibly transform the images and generate high-quality images.

## 5 CONCLUSION

In this paper, we propose an object-centric learning method, called **L**earning **O**bject-centric **R**epresentation from **M**ulti-viewpoint (LORM), to learn the viewpoint-dependent representation and viewpoint-independent representation from the multi-view scene in an unsupervised manner. We proposed a novel slot attention encoder to disentangle the scene attributes into viewpoint and object representations and a mixture patch decoder to reconstruct the image of the individual object with the corresponding viewpoint and object representations. The experimental results demonstrate that the proposed method performs more remarkably in decomposing scenes than the comparison methods and satisfactory disentanglement and viewpoint learning capability.

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

# A  DETAILS OF DATASETS

Configurations of the datasets used in this paper are presented in Table 2 and Table 3. CLEVR-A, SHOP, and GSO are synthetic 3D scene datasets. CLEVR-A and SHOP are generated based on the official code provided by (Johnson et al., 2017) and (Nazarczuk & Mikolajczyk, 2020), respectively. The size of the generated images of CLEVR-A and SHOP is $128 \times 128$. Code is modified to skip the check of object visibility because the observations of objects vary as viewpoints change. GSO is generated by selecting 10 kinds of 3D objects and 10 types of backgrounds to render the scene images with Kubric (Greff et al., 2022).

# B  DETAILS OF METRICS

Here we provide the calculation method of each metric, including 1) Adjusted Rand Index (ARI) (Hubert & Arabie, 1985) 2) Adjusted Mutual Information (AMI) (Vinh et al., 2009) 3) mean Intersection over Union (mIoU) 4) Mean Square Error (MSE) 5) Clustering Accuracy (ACC). In order to better describe the calculation, we define multiple variables here to facilitate use in the following subsections.

Suppose the test sets have $I$ visual scenes and each visual scene $V$ images from multiple unspecified viewpoints. let $\hat{K}_i$ be the true maximum number of objects appearing in the $i$th visual scene. and let $K_i$ be the estimated maximum number of objects appearing in the $i$th visual scene. Note that $\hat{K}_i$ and $K_i$ are not necessarily equal. $\hat{r}_i \in \{0,1\}^{V \times (\hat{K}_i+1) \times N}$ and $r_i \in \{0,1\}^{V \times (K_i+1) \times N}$ respectively represent the true and estimated one-hot vector of the $V$ viewpoints in the $i$th scene corresponding to the pixel-wise partitions (including the foreground and background). $\mathcal{D}_v^i$ denotes the index sets that belong to the object areas in the $t$th viewpoint of the $i$th scene, i.e., $\mathcal{D}_v^i = \{n \mid \boldsymbol{x}_{v,n}^i \in \text{object areas}\}$. Let $\hat{U}_{v,k}^i$ be the real index sets w.r.t. object $k$ in the $v$th viewpoint of the $i$th scene, i.e., $\hat{U}_{v,k}^i = \{n \mid \boldsymbol{x}_{v,n}^i \in \text{areas of object } k\}$ $(0 \leq k \leq \hat{K}_i)$. Let $U_{v,k}^i$ be the estimated index sets w.r.t. object $k$ in the $v$th viewpoint of the $i$th scene. $\hat{U}_{v,k}^i = \{n \mid \hat{\boldsymbol{x}}_{v,n}^i \in \text{areas of object } k\}$ $(0 \leq k \leq \hat{K}_i)$, where $\hat{\boldsymbol{x}}$ is the reconstructed image. Let $\hat{\boldsymbol{m}}^i \in [0,1]^{V \times \hat{K}_i \times N}$ and $\boldsymbol{m}^i \in [0,1]^{V \times K_i \times N}$ be the true and estimated pixel-wise masks that indicate the object(including the background) weight for each pixel in each viewpoint. Let $\hat{\boldsymbol{a}}^i \in [0,1]^{V \times \hat{K}_i \times N \times 3}$ and $\boldsymbol{a}^i \in [0,1]^{V \times K_i \times N \times 3}$ be the true and estimated appearance of objects in each viewpoint of the $i$th scene. Let $\hat{y}_k^i \in [0,...,C]$ denote the true label of the $k$th object in the $i$th visual scene, and correspondingly, $y_k^i \in [0,...,C]$ denotes the estimated label of the $k$th object in the $i$th visual scene, where $C$ denotes the category numbers, and the value greater than $C-1$ (i.e. $C$) indicates it is not an object.

## B.1  ADJUSTED RAND INDEX (ARI)

The computation of the Adjusted Rand Index (ARI) is described as:

$$\text{ARI} = \frac{1}{I} \sum_{i=1}^{I} \frac{b_{\text{all}}^i - b_{\text{row}}^i \cdot b_{\text{col}}^i / c^i}{\left(b_{\text{row}}^i + b_{\text{col}}^i\right)/2 - b_{\text{row}}^i \cdot b_{\text{col}}^i / c^i}. \tag{1}$$

In order to explain the meaning of each variable above in detail, $C(x,y)$ is used here to represent the combination number, i.e., $C(x,y) = \frac{x!}{(x-y)!y!}$; $v_{\hat{k},k}^i$ denotes the dot product, i.e., $v_{\hat{k},k}^i = \sum_{(v,n) \in \mathcal{S}} (\hat{r}_{v,k,n} \cdot r_{v,k,n})$, $b_{\text{row}}^i$, $b_{\text{col}}^i$ and $c^i$ in Eq 2 are described as:

$$b_{\text{all}}^i = \sum_{\hat{k}=0}^{\hat{K}_i} \sum_{k=0}^{K} C\left(v_{\hat{k},k}^i, 2\right), \tag{2}$$

$$b_{\text{row}}^i = \sum_{\hat{k}=0}^{\hat{K}_i} C\left(\sum_{k=0}^{K} v_{\hat{k},k}^i, 2\right), \tag{3}$$

$$b_{\text{col}}^i = \sum_{k=0}^{K} C\left(\sum_{\hat{k}=0}^{\hat{K}_i} v_{\hat{k},k}^i, 2\right), \tag{4}$$

$$c^i = C\left(\sum_{\hat{k}=0}^{\hat{K}_i} \sum_{(v,n) \in \mathcal{S}} \hat{r}_{v,\hat{k},n}^i, 2\right), \tag{5}$$

| Datasets | CLEVR-A | | | | SHOP | | | |
|---|---|---|---|---|---|---|---|---|
| Split | Train | Valid | Test | General | Train | Vaid | Test | General |
| # of Images | 5000 | 100 | 100 | 100 | 5000 | 100 | 100 | 100 |
| # of Objects | 3∼6 | 3∼6 | 3∼6 | 7∼10 | 3∼6 | 3∼6 | 3∼6 | 7∼ 10 |
| # of Views | 10 | | | | 10 | | | |
| # of Categories | 10 | | | | 10 | | | |
| # of Backgrounds | 1 | | | | 1 | | | |
| Image Size | 128×128 | | | | 128×128 | | | |
| Azimuth $\theta$ | [0,2$\pi$] | | | | [0,2$\pi$] | | | |
| Elevation $\rho$ | [10.5,12] | | | | [10.5,12] | | | |
| Distance $\phi$ | [0.15$\pi$,0.3$\pi$] | | | | [0.15$\pi$,0.3$\pi$] | | | |

Table 2: configuration of CLEVR-A and SHOP

| Datasets | GSO | | |
|---|---|---|---|
| Split | Train | Valid | Test |
| # of Images | 5000 | 100 | 100 |
| # of Objects | 3∼6 | 3∼6 | 3∼6 |
| # of Views | 10 | | |
| # of Categories | 10 | | |
| # of Backgrounds | 10 | | |
| Image Size | 128×128 | | |
| Azimuth $\theta$ | [0,2$\pi$] | | |
| Elevation $\rho$ | [7,9] | | |
| Distance $\phi$ | [0.35$\pi$,0.6$\pi$] | | |

Table 3: configuration of GSO

where $\mathcal{S} = \{1, 2, ..., V\} \times \{1, 2, ..., N\}$. When computing ARI-O, pixels in $\mathcal{S}$ that do not belong to objects will be removed, that is $\mathcal{S}' = \{1, 2, ..., V\} \times \{n \mid \boldsymbol{x}_n \in \text{objects areas}\}$; When ARI-A is calculated, all pixels in $\mathcal{S}$ will be used.

## B.2 ADJUSTED MUTUAL INFORMATION

The computation of Adjusted Mutual Information (AMI) is described as:

$$\text{AMI} = \frac{1}{I} \sum_{i=1}^{I} \sum_{t=1}^{V} \frac{\text{MI}(\hat{\boldsymbol{l}}^i, \boldsymbol{l}^i) - \mathbb{E}\big[\text{MI}(\hat{\boldsymbol{l}}^i, \boldsymbol{l}^i)\big]}{\big(\text{H}(\hat{\boldsymbol{l}}^i) + \text{H}(\boldsymbol{l}^i)\big)/2 - \mathbb{E}\big[\text{MI}(\hat{\boldsymbol{l}}^i, \boldsymbol{l}^i)\big]}, \tag{6}$$

where $\hat{\boldsymbol{l}}^i \in \mathbb{R}^{V \times (\hat{K}_i+1)}$. $\hat{\boldsymbol{l}}_v^i$ denotes the true probability distribution of the $t$th viewpoint in the $i$th visual scene, i.e., $\hat{\boldsymbol{l}}_v^i = \big\{|\hat{U}_{v,k}|/|\mathcal{D}_v^i| \mid 0 \le k \le \hat{K}_i\big\}$. $\boldsymbol{l}_v^i$ is the estimated probability distribution, i.e., $\boldsymbol{l}_v^i = \big\{|U_{v,k}|/|\mathcal{D}_v^i| \mid 0 \le k \le K_i\big\}$. H and MI respectively represent the entropy and mutual information of the distribution and their mathematical forms are described as:

$$\text{H}(\hat{\boldsymbol{l}}^i) = -\sum_{k=0}^{\hat{K}_i} \sum_{v=1}^{V} \hat{l}_{v,k}^i \log \hat{l}_{v,k}^i \tag{7}$$

$$\text{H}(\boldsymbol{l}^i) = -\sum_{k=0}^{K_i} \sum_{v=1}^{V} l_{v,k}^i \log l_{v,k}^i \tag{8}$$

$$\text{MI}(\hat{\boldsymbol{l}}^i, \boldsymbol{l}^i) = \sum_{m=0}^{\hat{K}_i} \sum_{n=0}^{K_i} \sum_{v=1}^{V} p_{v,m,n}^i \log \Big(\frac{p_{v,m,n}^i}{\hat{l}_{v,m}^i \cdot l_{v,n}^i}\Big), \tag{9}$$

where $\hat{l}_{t,k}^i$ and $l_{t,k}^i$ respectively note the true and estimated probability that the pixel in the $i$th image is partitioned to object $k$. $p_{v,m,n}^i$ denotes the probability w.r.t. pixels in the $v$th viewpoint of the $i$th scene are divided into objects $m$ in the first set and objects $n$ in the second set. $p_{t,m,n}^i$ is calculated

as follows:

$$p_{v,m,n}^i = \frac{o_{v,m,n}^i}{|\mathcal{D}_v^i|} = \frac{|\hat{U}_{v,m}^i \cap U_{v,n}^i|}{|\mathcal{D}_v^i|}. \tag{10}$$

The matrix $\boldsymbol{o}_v^i \in \mathbb{R}^{(\hat{K}_i+1) \times (K_i+1)}$ is called the contingency table. The expectation of MI can be analytically computed:

$$\mathbb{E}\big[\mathrm{MI}(\hat{\boldsymbol{l}}^i, \boldsymbol{l}^i)\big] = \sum_{v=1}^V \sum_{m=0}^{\hat{K}_i} \sum_{n=0}^{K_i} \sum_{k=(a_{v,m}^i+b_{v,n}^i-N)^+}^{\min(a_{v,m}^i, b_{v,n}^i)} \frac{k}{N} \cdot \log\left(\frac{N \times k}{a_{v,m}^i \times b_{v,n}^i}\right)$$
$$\frac{a_{v,m}^i! b_{v,n}^i! (N - a_{v,m}^i)! (N - b_{v,n}^i)}{N! k! (a_{v,m}^i - k)! (b_{v,n}^i - k)! (N - a_{v,m}^i - b_{v,n}^i + k)!}, \tag{11}$$

where $(a_{v,m}^i + b_{v,n}^i - N)^+ = \max(1, a_{v,m}^i + b_{v,n}^i - N)$, $a_{v,m}^i$ and $b_{v,n}^i$ respectively represent the sum of rows and columns w.r.t. $\boldsymbol{o}_v^i$:

$$a_{v,m}^i = \sum_{n=0}^{K_i} o_{v,m,n}^i, \quad b_{v,n}^i = \sum_{m=0}^{\hat{K}_i} o_{v,m,n}^i. \tag{12}$$

Similar to ARI calculation, when calculating AMI-O, we will only consider pixels belonging to the foreground, while AMI-A needs to consider all pixels.

### B.3 MEAN INTERSECTION OVER UNION

In order to compute mean Intersection over Union(mIoU), we need to do the object layer matching between the true and estimated masks.

$$\boldsymbol{\xi}^i = \mathrm{argmax}_{\boldsymbol{\xi}^i \in \boldsymbol{\Xi}^i} \sum_{v=1}^V \sum_{k=0}^{\hat{K}_i} \sum_{n=1}^N \hat{r}_{v,k,n}^i \cdot r_{v,\xi_k^i,n}^i, \tag{13}$$

where $\boldsymbol{\Xi}^i$ is the full arrangement of all entity(including the foreground and background) indexes in the $i$th visual scene. The computation of IoU is described as:

$$\mathrm{mIoU} = \frac{1}{I} \sum_{i=1}^I \frac{1}{\hat{K}_i} \sum_{k=1}^{\hat{K}_i} \frac{\sum_{v=1}^V \sum_{n=1}^N \min(\hat{m}_{v,k,n}^i, m_{v,k,n}^i)}{\sum_{v=1}^V \sum_{n=1}^N \max(\hat{m}_{v,k,n}^i, m_{v,k,n}^i)}. \tag{14}$$

### B.4 MEAN SQUARE ERROR

Suppose the reconstructed image of the $v$th viewpoint is described as $\hat{\boldsymbol{x}} \in \mathbb{R}^{V \times N \times 3}$, then the Mean Square Error (MSE) is computed as:

$$\mathrm{MSE} = \frac{1}{V} \sum_{v=1}^V \sum_{n=1}^N \|\boldsymbol{x}_{v,n} - \hat{\boldsymbol{x}}_{v,n}\|_2^2. \tag{15}$$

## C CHOICES OF HYPERPARAMETERS

**OCLOC** OCLOC was trained with the default hyperparameters described in the "exp_multi/config_blender_multi.yaml" and "exp_multi/config_kubric_multi.yaml" files of the official code repository[1] except the number of viewpoints was 6/6/6 on the CLEVR-A/SHOP/GSO dataset.

**SIMONe** SIMONe was trained with the default hyperparameters described in SIMONe (Kabra et al., 2021) except:1) the batch size was 4; 2)the episode length (ep_len) was 6; 3) the learning rate is $2 \times 10^{-4}$. During visualization, we chose 7 reconstructions of objects that have the largest masks and 1 reconstruction of the background manually for better comparison, even though the number of slots was 16 during training.

---

[1]`https://github.com/jinyangyuan/multiple-unspecified-viewpoints`

**STEVE**   The official STEVE implementation[2] was used. Models were trained with the hyperparameters described in the "train.py" file of the official code repository, except 1)the batch size was 6, and 2)the episode length (ep_len) was 6.

**DINOSAUR**   The official DINOSAUR implementation[3] was used. Models were trained with the hyperparameters described in the "configs/experiment/projects/bridging/dinosaur/movi _c_feat_rec.yaml" file of the official code repository, except the batch size was 64.

**LORM**   In all datasets, in order to avoid the neural network parameters falling into the local optimal, LORM was first trained with a single frame for 100 epochs and then trained with random 6 frames. The total number of epochs is 500. The batch size is 24/4/4 on the CLEVR-A/SHOP/GSO dataset. The initial learning rate is $3 \times 10^{-4}$,$1 \times 10^{-4}$ and $3 \times 10^{-4}$ for Image Encoder-Decoder(dVAE),Object-Centric Encoder and Mixture Patch Decoder respectively. Except for the learning rate of dVAE, they all decayed exponentially with a factor 0.5 every 250,000 steps after multiplying a parameter that increases linearly from 0 to 1 in the first 30000 steps.

The number of slots is 7/8/8 on the CLEVR-A/SHOP/GSO dataset in order to get the best performance. In the slot attention encoder, the number of iterations is 2. In the patch decoder, the number of patches is 1024. The size of view representations is 4 on all datasets and the size of object representations is 64/128/64 on the CLEVR-A/SHOP/GSO dataset.

## D   EXTRA EXPERIMENTAL RESULTS ON THE SHOP AND GSO DATASETS

The qualitative comparison of the proposed LORM, STEVE, DINOSAUR, SIMONe and OCLOC on the CLEVR, SHOP and GSO datasets is shown in Figures $6 \sim 10$, respectively.

As shown in Figures 6, the proposed method as well as STEVE can segment and reconstruct scenes well on all datasets.DINOSAUR can also have great segmentations, but it can not reconstruct it learns object representations from feature space. SIMONe and OCLOC performed well in simple datasets such as CLEVR-A and SHOP, but they all failed in complex dataset(GSO)

---

[2]https://github.com/singhgautam/steve
[3]https://github.com/amazon-science/object-centric-learning-framework

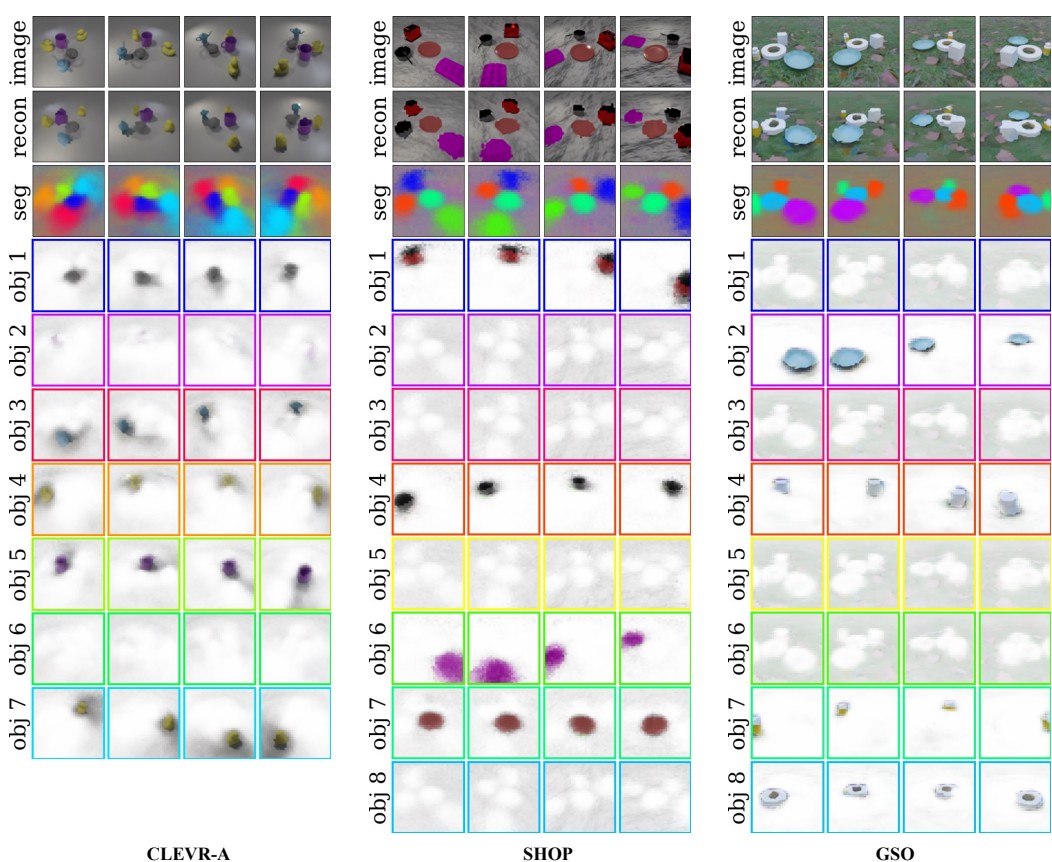

Figure 6: The Decomposition results of LORM on three datasets.

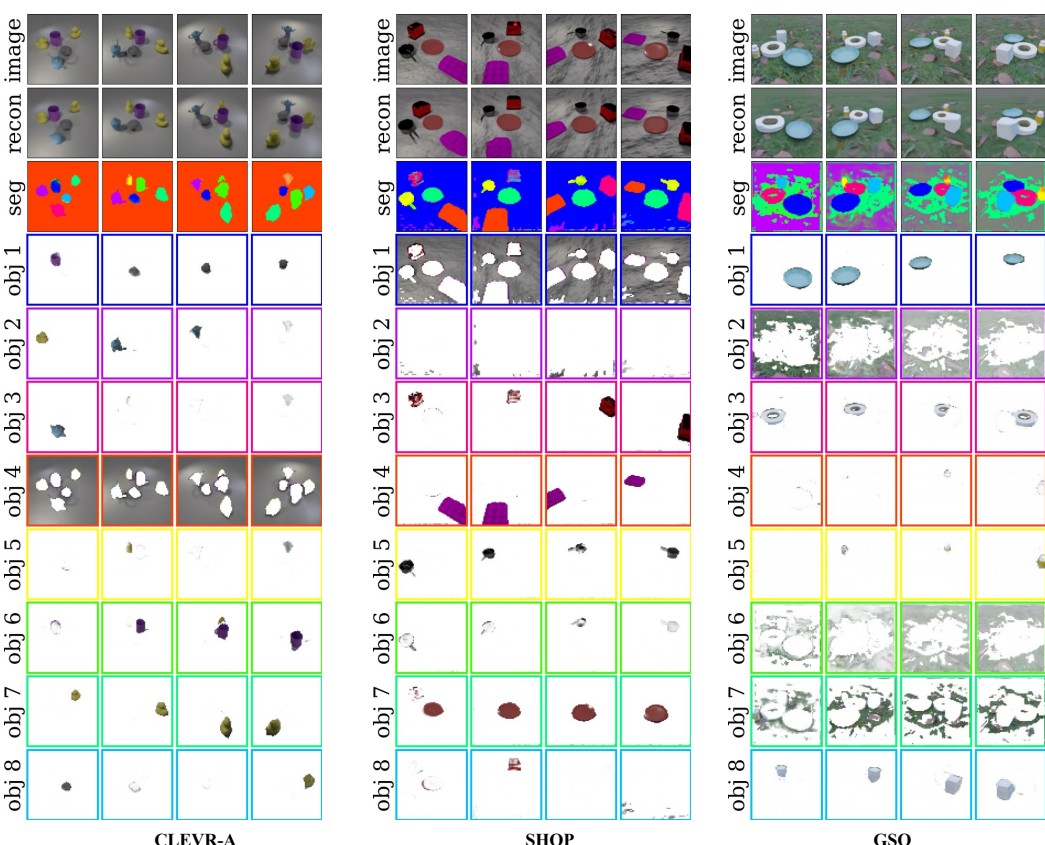

Figure 7: The Decomposition results of STEVE on three datasets.

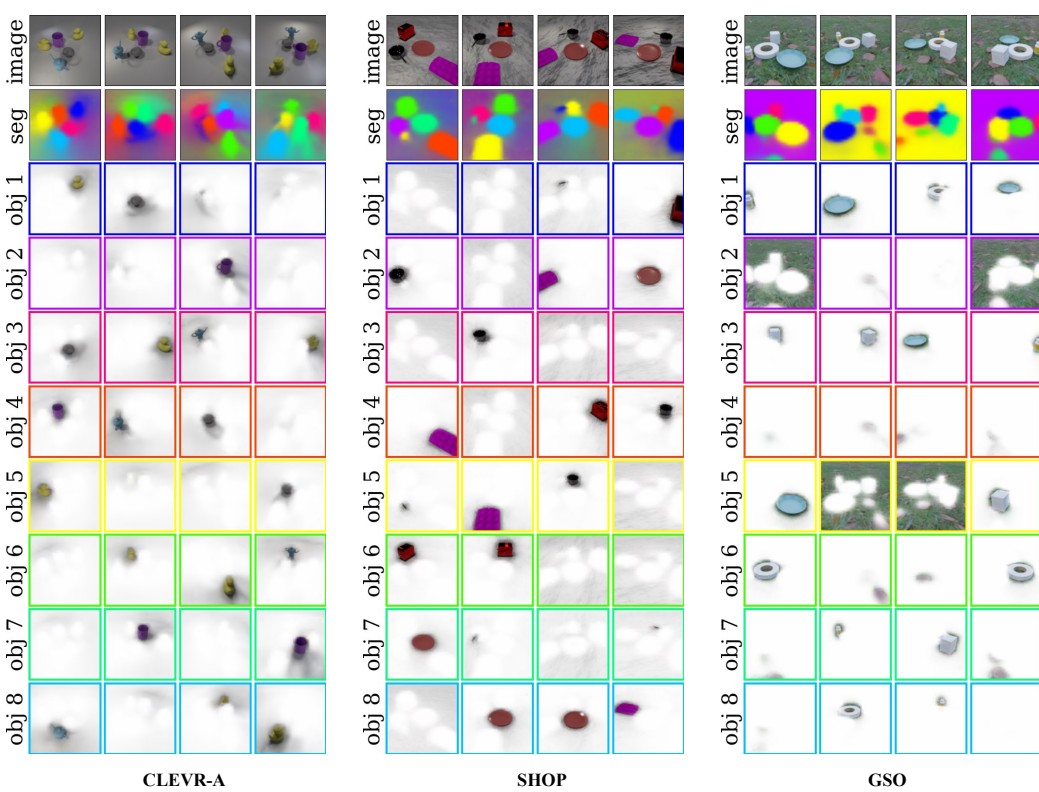

Figure 8: The Decomposition results of DINOSAUR on three datasets.

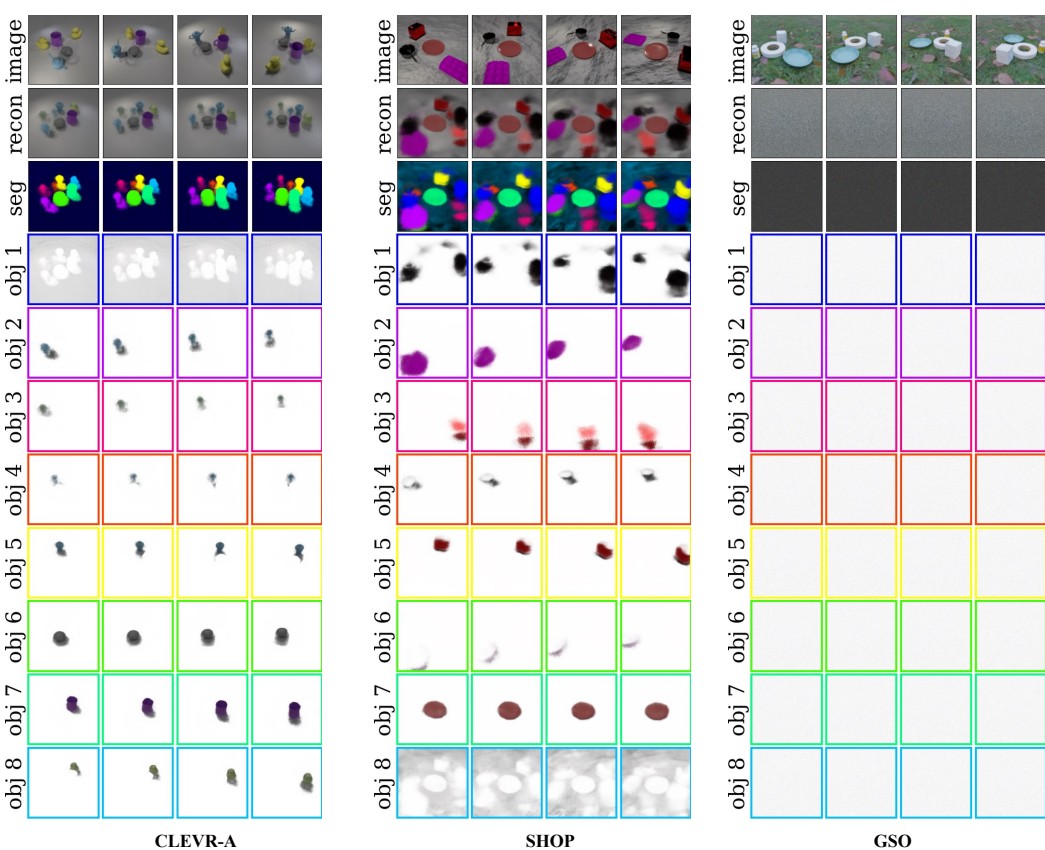

Figure 9: The Decomposition results of SIMONe on three datasets.

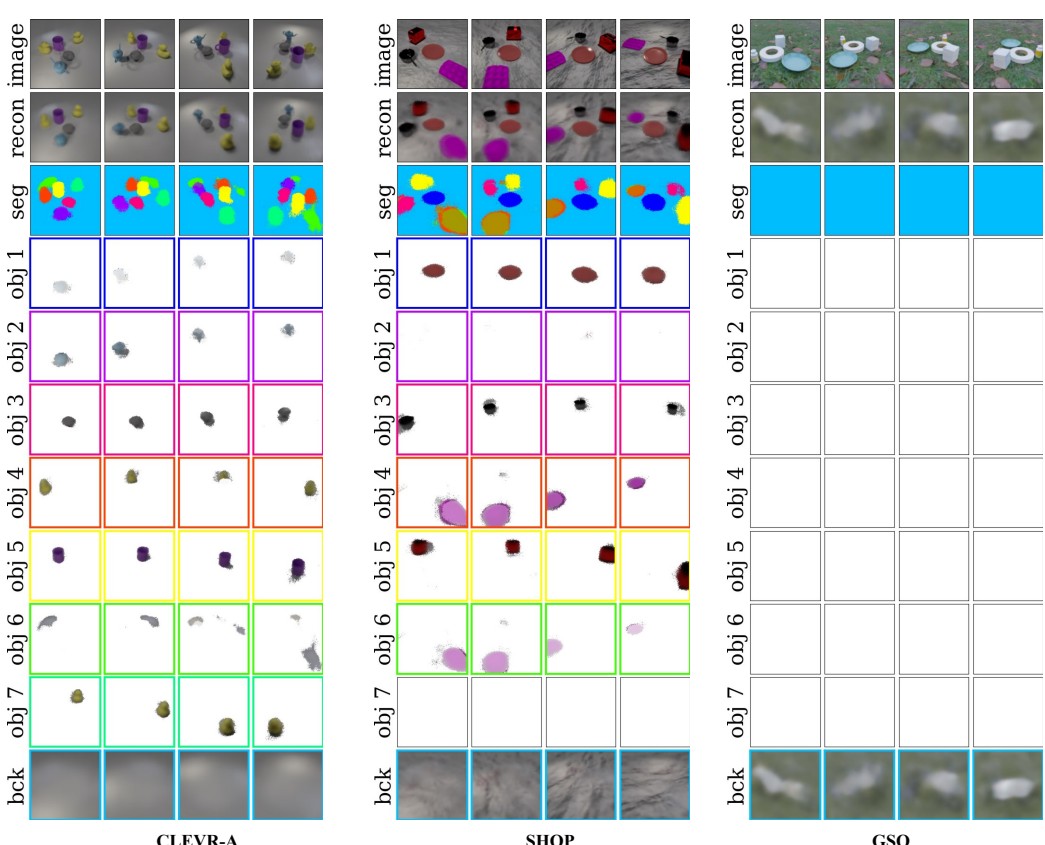

Figure 10: The Decomposition results of OCLOC on three datasets.

