# OpenReview forum: "Unsupervised Learning of Object-Centric Representation from Multi-Viewpoint Scenes"
_ICLR.cc/2024/Conference — ICLR 2024 Conference Withdrawn Submission_

### Official Review · Reviewer_LpCd · 2023-10-31

**Soundness:** 3 good
**Presentation:** 2 fair
**Contribution:** 2 fair
**Rating:** 3
**Confidence:** 4

**Summary:**

this paper investigates the problem of object-centric representation learning. specifically, it focuses on the multi-view setup, and extends existing work SIMONe and STEVE. the full system is evaluated on multiple benchmarks.

**Strengths:**

+ leanring object-centric representation from multiple views is a interesting problem
+ clear figures
+ quantatitive and visual results

**Weaknesses:**

- the proposed object-centric encoder (section 3.1) seems like a modified version of SIMONe, where the randomly sample temporal representation in SIMONe is replaced by CNN features in the proposed architecture. for the manuscript to better convince the audience that this is in fact a good choice, it would be better to include variant study that directly compares this design against that of SIMONe.
- the decoder and reconstruction loss is heavily inspired by STEVE. since it is not very clearly introduced (see point below), the reviewer suspects (from euqations on P5) that the main difference is to consider the weighted sum of multiple features. again, like the point above, it takes direct experimental comparisons to convince the audience this is indeed a better choice, yet the reviewer sees no ablation/variant study in the current manuscript.
- for a method that is heavily inspired by previous works SIMONe and STEVE, the reviewer would expect it to outperform both by taking advantages from their specific designs. however, as indicated in Table 1, oftentimes the proposed method fall behind existing methods. the reviewer would expect more justification on this. is it because of some issue that no one has spotted before? or is it just a indication of unsatisfactory performance?
- the defination of $u$ is rather unclear. is it the attention weight? what is the difference between $u$ and $\tilde{a}$?
- the introduction of the mixture patch decoder is not clear.

-- minor: $\tilde{a}$ in P4 bottom

**Questions:**

see weakness

---

### Official Review · Reviewer_AcUg · 2023-11-01

**Soundness:** 2 fair
**Presentation:** 1 poor
**Contribution:** 2 fair
**Rating:** 3
**Confidence:** 3

**Summary:**

This paper presents a self-supervised slot-based method for decomposing a multi-view scene into objects. The method relies on encoding the viewpoints individually, and then conditioning a slot attention encoder based on the image representations, then decoding these objects with a patch-wise technique (not entirely clear to me), followed by re-assembling the components to form a reconstruction of the images.

**Strengths:**

The results are difficult to judge, and I think quite often they are qualitatively bad (see Figure 2), but this is an open area where I think many people are struggling to make progress, and breakthroughs here would be exciting for many.

**Weaknesses:**

Overall this paper's most obvious weakness is its presentation. It is very difficult to understand what exactly the authors did.

The method seems extremely complex, or it is poorly described. Multiple paragraphs in the method read more like code than English. It would be great to provide a mid-level description that expresses why certain design choices were made, and what each component is trying to achieve.

The second paragraph of the intro is very hard to read. Besides appearing like a list of acronyms with citations, the sentences which begin with "The methods," have incorrect usage of commas.

The paper says multiple times that the method will "disentangle the scene representation into viewpoint and object representation". Shouldn't there be multiple objects, and also perhaps a background? A camera pose and an "object representation" does not really achieve the compositional structure hinted at in other parts of the work. Judging by the other parts of the paper, my current guess is that sentences like these are simply inaccurate; perhaps they came from some earlier draft where the method was different?

The paper says that unlike other work, Slot Attention Encoder is input multiple viewpoints simultaneously. I think actually OSRT did this too (Sajjadi et al., 2022).

The last paragraph of 3.1 is difficult to read. It would be better if none of the sentences started with math symbols. Multiple sentences have problems with plurals (e.g., "The attention map are").

The paper says "One of the object-centric learning methods’ essential and fundamental abilities is reconstructing the image of individual objects in the scene". I am not convinced by this. Detectors are object-centric learning methods, and they do not typically have a reconstruction objective.

The paper says that "To decompose complex natural scenes, DINOSAUR reconstructs the patch features extracted by a pre-trained self-supervised transformer (Caron et al., 2021), while STEVE reconstructs the whole scenes through the DVAE decoder. Therefore, we propose... " It would be good to have a citation for STEVE here. Also the sentence makes it sound like DINOSAUR depends on STEVE. Finally, I do not follow the reasoning implied by the "Therefore".

The paper says "Multiple fully connected layers are used to consist of gpatchdec to reduce the complexity of the model construct as much as possible." What is meant by "the model construct"? How does using multiple fully connected layers reduce complexity of something?

I could not figure out what the connection is between the image encoder-decoder and the rest of the pipeline. I see that there is a cross entropy loss between "o" and "z", but I do not know why this is here. The paper mentions "a categorical distribution with N classes" a few times but I could not find more information about this. What is N? What are the classes? Why do we want to classify image patches according to these classes? What does this achieve?

In general I am not sure about the results. The paper describes the results as "satisfactory" and "gratifying", but this is entirely subjective. The quantitative results do not seem to show a clear win for the LORM method. Others, like DINOSAUR and STEVE seem to do better on the considered datasets. I see that DINOSAUR uses a pre-trained component, so I wonder if maybe another version of LORM might be good to try: one with a similar pre-trained part (i.e., semi-supervised setting).

Objec-Centric -> Object-Centric

**Questions:**

Is it possible to try OSRT as a baseline here?

Are camera poses used anywhere? I could not find them anywhere in the paper, but for a multi-view method it may make sense.

---

### Official Review · Reviewer_UJby · 2023-11-01

**Soundness:** 3 good
**Presentation:** 3 good
**Contribution:** 2 fair
**Rating:** 5
**Confidence:** 4

**Summary:**

The paper proposes a method to explicitly disentangle camera-specific and camera-invariant features in object-centic  learning methods that model multi-view data. They propose do so by having seperate encoders for view-dependent and view-independent decoding while training unsupervised for image reconstruction task

**Strengths:**

- The paper takes a step towards building models that can disentangle camera-depdent and independent representation.
-  The paper does dense evaluation wrt benchmarks and baselines.
- The paper shows interesting results for editting two images and also interpolation between different viewpoint

**Weaknesses:**

- The paper does a good qualitative job however doesn't show any quantiative result showing that disentanglement indeed happens. Further i couldn't find any ablations which show the benefit of having viewpoint seperation vs not.
-  No downstream performance result, it would be helpful to see if such a representation is helpful in learning downstream tasks that are invariant of equivariant to the viewpoint.
- The work gets poor results on metrics except AMI-O and ARI-O. It's unclear why or when should one judge performance based on ARI-O/AMI-O instead of the other evaluation metrics in the paper.
- Can the paper explain the trade-offs of using a SSL representation like DINOSAUR vs not. Becoz in all the evaluation metrics DINOSAUR seems to outpeform the current method.

**Questions:**

Please answer the questions mentioned in the weakness section above.